# Extravascular Ultrasound (EVUS) to Assess the Results of Peripheral Endovascular Procedures

**DOI:** 10.3390/diagnostics13071356

**Published:** 2023-04-06

**Authors:** Stefano Fazzini, Federico Francisco Pennetta, Valerio Turriziani, Simona Vona, Andrea Ascoli Marchetti, Arnaldo Ippoliti

**Affiliations:** Vascular and Endovascular Surgery, Department of Biomedicine and Prevention, Tor Vergata University of Rome, Via Cracovia, 50, 00133 Roma, Italy

**Keywords:** duplex ultrasound, PAD, angiography, EVUS, lithotripsy

## Abstract

Contrast arteriography (CA) is considered the gold standard to evaluate any phase in peripheral arterial disease (PAD) interventions, from diagnostics to final results. Nevertheless, duplex ultrasonography (DUS) mostly used for the pre/postoperative phase and follow-up control, could be a potential intraoperative adjunctive imaging tool to assess the effects of endovascular revascularization in patients with iliac and femoropopliteal lesions. The PAD “duplex-assisted” protocol includes a preoperative DUS control followed by an intraoperative and a postoperative control. The most important parameters are pulsed doppler spectral analysis and waveform changes, which are impossible to detect with intravascular ultrasound (IVUS). By using a similar acronym, the intraoperative DUS has been previously described as extravascular ultrasound (EVUS). B-mode imaging, color flow, and peak systolic velocity (PSV) are considered. EVUS could be very useful to evaluate the effects of endovascular treatment, mainly in cases of unclear CAs, severe calcifications and/or dissections. In the context of the “leaving nothing behind” strategy, EVUS can drive the physician to evaluate the absence of flow-limiting dissections and decide which target lesion should be treated with antirestenotic therapy, further vessel preparation, or stenting. The EVUS protocol could be a safe and feasible option to improve the completion assessment of endovascular PAD treatment. A better ultrasound waveform is a sign of improved luminal gain and compliance, which is extremely important to finalize the results of new peripheral device technology, such as intravascular lithotripsy.

## 1. Introduction

The use of preoperative and postoperative DUS is renowned for being a safe and feasible option in the scope of diagnostics and follow-up of PAD.

Existing studies have demonstrated the reliability of DUS in comparison with CA. Extensive and encouraging data about its sensitivity and sensibility are available [1,2].

Massive technology improvements have led to duplex scanners with higher definition B-mode images and ameliorated color flow overcoming unfavorable imaging obtained with inadequate, antiquated technology [3,4].

Despite DUS being a widely available device, it is barely used, and scarce reports about its intraoperative use are present, such as in renal surgical revascularization [5] and in carotid artery stenting [6].

Previous literature examining DUS focused mainly on comparison with CA [1,2,3,4], but no data are available about their combined use.

Intraoperative DUS could be an optimal adjunctive imaging technique in endovascular interventions to evaluate the immediate results, mainly in case of unclear CA.

Endovascular treatments have been recently used even in very complex peripheral diseases, such as extensive and severe calcifications, chronic total occlusions (CTO), small vessels (common in female gender), and critical limb-threatening ischemia (CLTI). In this context, the “leaving nothing behind” strategy is extremely difficult to reach, and a correct evaluation of CA imaging can be difficult. A duplex-assisted approach could be useful to confirm or not the immediate results of CA and drive physicians in the choice of further treatments.

A new duplex-assisted protocol is proposed in patients undergoing peripheral endovascular treatment.

## 2. Technical Aspects, The “EVUS” Protocol

The EVUS (extravascular ultrasound) protocol can be applied to patients undergoing endovascular repair of peripheral lesions.

Operatory position (Figure 1) is supine with a sterile field extending from the hip to the feet divided into three different spots upon the projections of the femoral (groin), popliteal (popliteal fossa), and tibial arteries (ankle) to check the outflow of iliac, femoropopliteal and tibial arteries, respectively.

### 2.1. Preoperative Dus

The first DUS exam is performed in the preoperatory phase. The patient is already in the aforementioned position in the operatory room.

The sequence of DUS starts by imaging the inflow artery and proceeding downstream. Sagittal and transverse imaging of the intended repair sites is obtained. The scanning technique employs B-mode, color doppler imaging, pulsed doppler spectral analysis, and measurement of PSV. In the pulsed doppler, waveform shapes are considered. A multiphasic waveform (triphasic, biphasic) crosses the zero-flow baseline and contains both forward and reverse velocity components. A monophasic waveform does not cross the zero-flow baseline throughout any part of the cardiac cycle [7].

PSV is then considered: PSV > 200 cm/s and a velocity ratio (Vr) > 2.5 indicates a ≥70% stenosis of the target lesion.

A linear 7-4 MHz probe is generally used. This protocol takes into account mainly outflow vessels distal to the target lesion, common femoral artery, and popliteal artery for the iliac and femoral district, respectively.

Probes are covered with a sterile sheath. In the exemplifying cases presented duplex scan was performed with an Esaote Mylab30 (Esaote, Genova, Italy).

In case of significant stenosis or CTO, a duplex scan usually shows a monophasic pattern of the pulsed doppler spectral analysis distal to the target lesion.

Furthermore, a triphasic waveform is rarely detected in elderly patients with very diffuse atherosclerotic disease, dialysis, or cases with a previous procedure (e.g., stenting), due to the lack of elasticity in vessel walls. In such cases, a preoperative mono-phasic waveform progressing into a bi-phasic postoperative waveform should represent a technical success (case 1–2).

Subsequently, the endovascular repair, generally defined as any procedure performed to restore a better/optimal luminal gain, is performed as planned, based on lesions and symptoms of the patients.

Complex peripheral endovascular treatments can be performed with multiple devices and techniques that are mainly divided into two phases, depending on different target lesion characteristics (stenosis, occlusion, length, presence of calcium, and severity of calcifications) and districts (aorto-iliac, femoropopliteal or tibial):Vessel preparation: Plain old balloon angioplasty (POBA), scoring balloons, atherectomy devices, and intravascular lithotripsy (IVL).Completion or primary treatments: antirestenotic therapy (drug eluting balloons, DEB) and/or stenting (bare metal and/or covered stents, drug eluting stents and/or mimetic stents).

### 2.2. Intraoperative Dus (EVUS)

EVUS is therefore performed. Detailed images of the repair site are obtained. DUS is performed in the same setting as the preoperatory one. B-mode and color doppler imaging are employed to estimate the presence of residual stenosis and dissections. A residual stenosis < 30% is considered a technical success. Dissections are treated only if flow-limiting (D-E-F) [8,9].

Pulsed doppler spectral analysis and PSV are considered. The presence of a multiphasic waveform or the absence of PSV alterations suggests procedural success. In the absence of stenting, a waiting time of 15 min is recommended to detect early recoil, a potential risk in case of concentric/severe calcifications [10].

A postprocedural EVUS with a monophasic waveform or with PSV alterations indicates the need for further treatment (e.g., different or more aggressive vessel preparation and/or stenting). EVUS should then be repeated to assess the efficacy of the adjunctive treatment. In patients with diffuse atherosclerosis, with no significant multiple stenosis or small vessels, an increased monophasic waveform could be considered a technical success (case 4).

### 2.3. Postoperative Dus and Follow-Up

Postoperatory DUS is performed at 24h and then used as surveillance protocol at 1, 3, 6, and 12 months and yearly thereafter. Ankle-brachial Index (ABI) is always performed in the pre-postoperative phase and at follow-up controls. Primary patency, defined as a target lesion without severe recoil, restenosis, occlusion, and target lesion reintervention, is assessed.

EVUS protocol is represented in Figure 2.

We describe the technique in two main districts: Iliac (cases 1 and 2) and femoral-popliteal lesions (cases 3 and 4). Informed consent was obtained from all subjects involved in the study.

## 3. EVUS in Iliac Treatment

**Case 1.** The patient was a 74 year-old male with a preocclusive calcified lesion of the proximal right external iliac artery and an occluded superficial femoral artery. He referred 20 m walking distance and nocturnal rest pain of the right leg (Rutherford category 3/4) [8] with an ABI of 0.45; he was previously treated by an endovascular abdominal aortic aneurysm repair (EVAR) therefore, contralateral femoral access was not feasible. A percutaneous left brachial access with 6 Fr introducer sheath was performed to avoid the puncture of an extremely calcified right common femoral artery (~50% stenosis). In this case, the right groin was examinable by DUS during the entire procedure in the absence of any introducer sheath. The preoperative DUS showed a monophasic waveform at the level of the common femoral artery Figure 3E).

After predilatation with a 3 mm POBA, 10 cycles of lithotripsy were delivered at the target lesion with IVL M5+ 7 mm catheter (Shockwave Medical, Santa Clara, CA, USA).

Due to the extensive vessel calcifications, the completion angiogram showed unclear imaging (good flow in terms of contrast medium velocity with a suboptimal arterial vessel filling) despite the use of two different C-arm projections. EVUS showed an optimal result with monophasic flow changing to multiphasic (Figure 3F). Neither DCB, nor stenting was necessary. The postoperative ABI was 0.8 with a walking distance > 1 km; therefore, any procedure of the superficial femoral artery was not required. The patient was discharged with double anti-platelet therapy (DAPT) for three months, followed by lifelong single anti-platelet therapy with 100 mg of cardioaspirin and statins. Primary patency was maintained at follow-up DUS controls, with the same waveform for up to 18 months.

In this case, EVUS clarified doubts about different angiograms and represented the final assessment to avoid stenting of a calcific plaque close to the hypogastric artery, potentially at risk for complications.

**Case 2.** The patient was a 61 year-old female with a calcified aortic bifurcation and a tight stenosis of the right common iliac artery, while a mild stenosis was present in the contralateral side (Figure 4A). Walking range was <50 m in the right side (Rutherford category 3) [8] with ABI of 0.6 and 1 in the right and left sides, respectively. The preoperative DUS showed a monophasic waveform on the right common femoral artery and a triphasic waveform on the left side (Figure 4D).

Percutaneous femoral access with 7 Fr introducer sheath was performed. Intra-arterial pressure measurement showed a 22 mmHg gradient between the aorta and right common femoral artery. Six cycles of lithotripsy were then delivered at the target lesion with IVL M5^+^ 8 mm. Completion angiogram showed optimal recanalization of the occlusion with 20% residual stenosis (angiographic assessment) and a pressure gradient of 2 mmHg. EVUS showed an optimal result with multiphasic flow in the right external iliac and common femoral artery (Figure 4E).

The postoperative ABI was 1 with an unlimited walking distance bilaterally. The patient was discharged with double anti-platelet therapy (DAPT) for three months followed by lifelong single anti-platelet therapy with 100 mg of cardioaspirin and statins. Primary patency was maintained at follow-up DUS controls, with optimal multiphasic waveform at 3 months.

In this case, the IVL stand-alone treatment together with EVUS allowed a single-access procedure to focus on the spot lesion at the origin of the right common iliac artery in place of a kissing stent; this option, though with proven outcomes, could be potentially considered an overtreatment in case of the absence of severe disease on the contralateral axis, leaving the aortic bifurcation free of stents for future procedures (e.g., crossover access).

## 4. EVUS in Femoropopliteal Treatment

**Case 3.** The patient was a 74 year-old male with a short (5 cm) calcific occlusion of the right superficial femoral artery. Walking range was <100 m (Rutherford category 3) [11] with ABI of 0.6. The preoperative DUS showed a monophasic waveform at the level of the right popliteal artery (Figure 5B,C).

Through a 6Fr left femoral access, a predilatation with POBA 3 mm and 10 cycles of lithotripsy were performed (IVL M5 + 6 mm). An antirestenotic treatment with 6 mm DCB was used. After the procedure, luminal gain was completely restored, and the CA showed a vessel profile with few irregularities and very good flow.

EVUS showed the absence of residual stenosis and/or dissection at the level of the target lesion, and a multiphasic flow was detected at the level of the popliteal artery (Figure 5H,I). The postoperative ABI was 1 with an unlimited walking distance. The patient was discharged with double anti-platelet therapy (DAPT) for three months, followed by lifelong single anti-platelet therapy with 100 mg of cardioaspirin and statins. Primary patency was maintained at follow-up DUS controls for up to 12 months.

In this case, EVUS gave us a better completion assessment compared with CA to avoid stenting of a distal superficial femoral artery and finalizing the procedure with DCB.

**Case 4.** The patient was a 72 year-old female with a short occlusion (<5 cm) of the right superficial femoral artery in the middle-distal section. Walking range was <50 m (Rutherford category 3) with an ABI of 0.6.

The preoperative DUS showed a direct/suboptimal inflow (proximal superficial femoral artery with good monophasic waveform, a sign of a small vessel with multiple stenosis) and a poor outflow (popliteal artery with a very low monophasic waveform) (Figure 6A–C).

Through left 4 Fr transfemoral access, after an intraluminal antegrade recanalization, a vessel preparation with POBA (2-3-4 mm balloons) and completion 4 mm DCB were performed. An unclear dissection (type C/D) was present, but the EVUS showed a better monophasic flow at the level of popliteal artery (fully comparable with the inflow) and the absence of significant flow-limiting dissections/residual stenosis (Figure 6E,F). In this case, considering the proximal extension of the pathology, no further treatment and/or stenting have been performed. On the first postoperative day, thanks to the better compliance and the vessel remodeling after the recanalization, the waveforms showed a further improvement with triphasic flow in the popliteal and tibial artery (Figure 6G).

The postoperative ABI was 1 with an unlimited walking distance. The patient was discharged with double anti-platelet therapy (DAPT) for three months, followed by lifelong single anti-platelet therapy with 100 mg of cardioaspirin and statins. Primary patency was maintained at follow-up DUS controls for up to 3 months.

In this case, considering the high risk of long stenting in a narrowed and diffused diseased superficial femoral artery, EVUS allowed us to detect a better waveform and avoid further treatments.

## 5. Discussion

In the era of duplex-assisted arterial access, portable ultrasound machines have become widely available in the operatory room. Therefore, different and further intraoperative use of this kind of vascular imaging could be substantially increased.

EVUS has some important adjunctive advantages beyond helping the surgeon during the endovascular procedure. First of all, a detailed evaluation of arterial diameters could favor the proper sizing of the materials, such as balloons and/or stents. Short lengths and characteristics of the lesions are more precisely evaluated.

Both these features can be achieved with IVUS. However, such a method is invasive, expensive, and time-consuming. Moreover, no pulsed doppler spectral analysis can be obtained.

CA is considered as the current gold standard in peripheral interventions for all phases of the procedure, including the final assessment.

Widening in the application of endovascular techniques to complex peripheral diseases (severe calcifications, CTO, small vessels, and CLI) and new devices such as IVL could lead to suboptimal angiograms. CA misinterpretation may therefore lead to unnecessary stenting or adjunctive treatment.

The use of a duplex scan could assist in an informed decision on whether to stent or not since suboptimal angiograms (similar to dissections, without a linear contrast-enhanced vessel wall) could be present even if clinically uneventful [8,9].

Moreover, the perfect CA at any cost could lead to overtreatments, whereas EVUS could limit further treatments in many cases as we described. The absence of significant residual lesions and/or flow-limiting dissection at the target vessel or a better ultrasound waveform in the outflow vessel means a better luminal gain and compliance leading to a -better walking distance or Rutherford category.

In our protocol (Figure 2), ultrasounds are used in all phases of the clinical setting, from preoperative to intra (EVUS) and postoperative assessment.

The effectiveness of ultrasound imaging can be potentially very high and reproducible, provided that skilled operators know standard and complex cases as well.

The arterial stiffness in some patients (dialysis, long stenting, diffused calcifications, narrowed vessels) may present with monophasic waveforms without significant stenosis.

On the other hand, a better monophasic waveform (case 4) could represent the completion assessment for some patients.

In the presence of doubt about the hemodynamic significance of residual stenosis, an intra-arterial pressure measurement across the stenosis could be performed in the iliac district (TASC II) [12]: If there is a low (<10 mmHg) pressure gradient across the stenosis further procedures may not be necessary.

In addition, multiple angiograms and the use of oblique C-arm projections could potentially lead to a higher amount of iodinated contrast-medium and fluoroscopy dose.

The EVUS protocol could reduce these complications and be used in patients with chronic renal failure as an alternative strategy to reduce contrast agent use, such as with carbon dioxide contrast.

Many different commentaries have been published on the “leave nothing behind” strategy [13,14]. Such a strategy consists in avoiding stenting or implantable materials as far as possible. The approach consists in using different techniques to perform the ideal vessel preparation for antirestenotic therapy, mainly in the femoro-popliteal district: angioplasty, debulking, or intravascular lithotripsy are used in vessels with stenosis, chronic total occlusions, or extensive calcifications, respectively. Stenting should be reserved for cases with flow-limiting dissections or with residual stenosis >50%. The possibility of avoiding stenting is a real benefit in “no-stent zones”, such as popliteal, distal superficial (cases 3 and 4), or common femoral artery, where different mechanical forces could be at high risk of stent occlusion and/or fracture. The stent patency has proven outcomes in the iliac district, but also in this area, there are different conditions in which avoiding or limiting the number of stents could be useful. The aortic or iliac bifurcation are some of these sites (case 1 and 2).

The goal of changing vessel compliance is to restore natural mechanics to the vessel, adding elasticity and increasing pulsatility. Compliance is key and opens the door to successful treatments, changing previous procedure plans and preserving many future options. EVUS protocol could help to assess intraoperatively the increased vessel compliance and avoid unnecessary overtreatment, and this is extremely useful if we are using intravascular lithotripsy (case 1, 2, and 3).

The IVL cracks calcium and changes its structure, increasing luminal gain and vessel compliance at the same time because of transverse and longitudinal fractures. For this reason, the angiographic vessel profile could be unclear, and these fractures may seem like dissections or different grey areas inside a contrast-enhanced vessel.

Even if calcium is an obstacle for ultrasounds, the vessel flow is easily detected distal to the target lesions. For example, IVUS is extremely precise in detecting luminal gain and the EVUS gives us most of the information we need after any peripheral procedure, among which there is a better compliance.

Even EVUS can have some limitations due to the use of old equipment, lack of experience and/or a standardized exam protocol. Furthermore, extremely calcified arteries (shadow cones), long lesions, and skin quality problems such as the presence of wounds, ulcers and/or heavy scarring can affect the accuracy of the exam. The use of a linear probe could potentially be feasible for most of the treatments, considering the outflow of treated vessels are checked (iliac, above and below the knee vessels detected at the groin, popliteal fossa, and ankle, respectively). Sometimes, the presence of the sheath could be a drawback to assessing the flow, especially in the case of very narrowed vessels.

With this simple, cost-effective protocol, endovascular treatment of peripheral lesions could be tailored to the patient’s specific condition.

## 6. Conclusions

The EVUS protocol is an appropriate technique together with angiographic images for intraoperative assessment of arterial revascularization. EVUS enables the physician to understand the hemodynamics of residual stenosis and the need for further treatment.

A duplex-assisted endovascular procedure is extremely useful in the case of calcific diseases where the assessment of the final result is not restricted to the best and ideal angiogram but also to evaluate better vessel compliance, which is well detected by a better doppler waveform.

In this context, new devices such as Intravascular Lithotripsy (IVL) could favorably match the use of EVUS to elevate the vessel preparation phase and enable new vascular pathways.

## Figures and Tables

**Figure 1 diagnostics-13-01356-f001:**
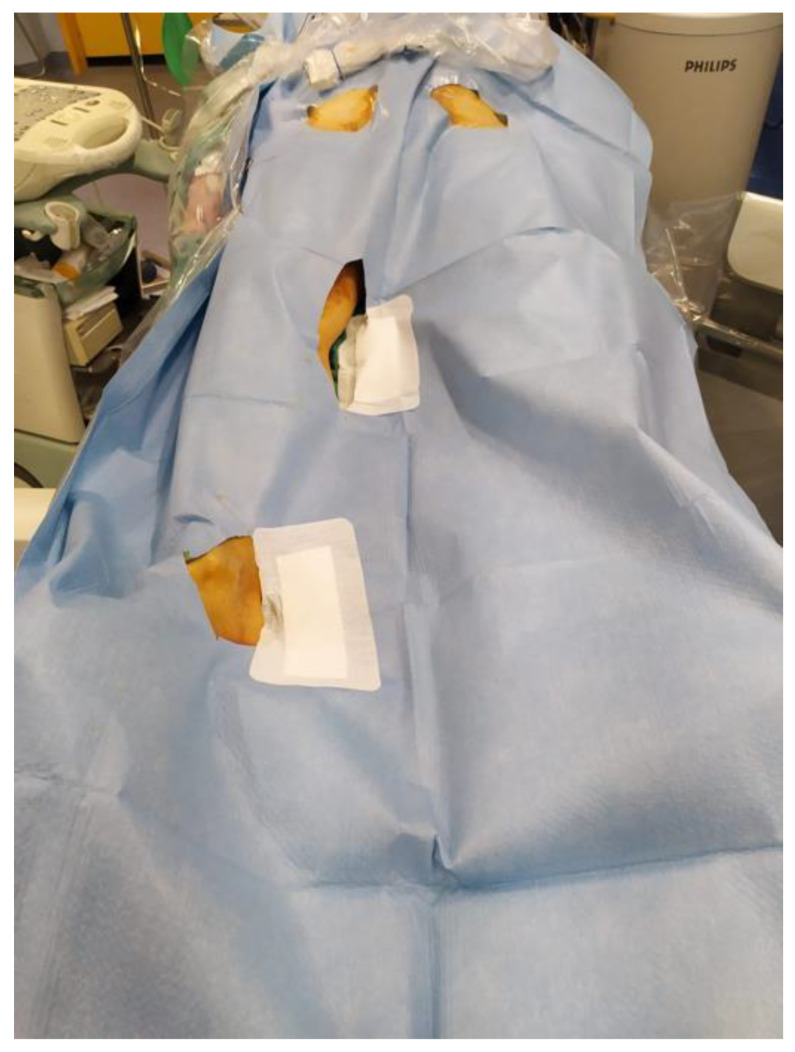
EVUS (extravascular ultrasound): Sterile setting in the operatory room before a peripheral endovascular procedure (occlusion of the right superficial femoral artery) with three different spots upon the projections of femoral, popliteal, and tibial arteries. The leg is prepared with a double sterile drape to approach any leg spots during the procedure.

**Figure 2 diagnostics-13-01356-f002:**
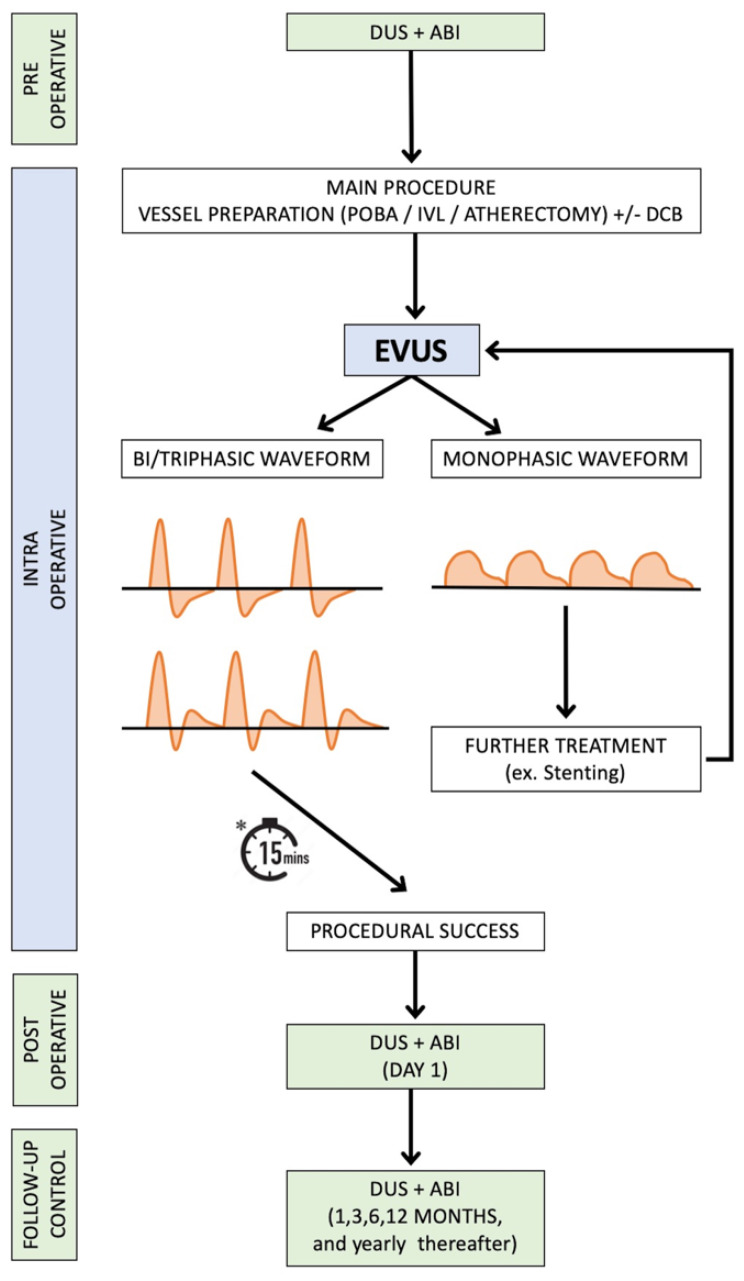
Diagnostic algorithm for peripheral arterial endovascular treatments and EVUS protocol. * In the absence of stenting, a waiting time of 15 min should be considered mandatory in case of heavily calcified plaque at high risk of recoil [10]. DUS, duplex ultrasound; ABI, ankle-brachial index; EVUS, extravascular ultrasound; POBA, plain old balloon angioplasty; IVL, intravascular lithotripsy; DCB, drug-coated balloon.

**Figure 3 diagnostics-13-01356-f003:**
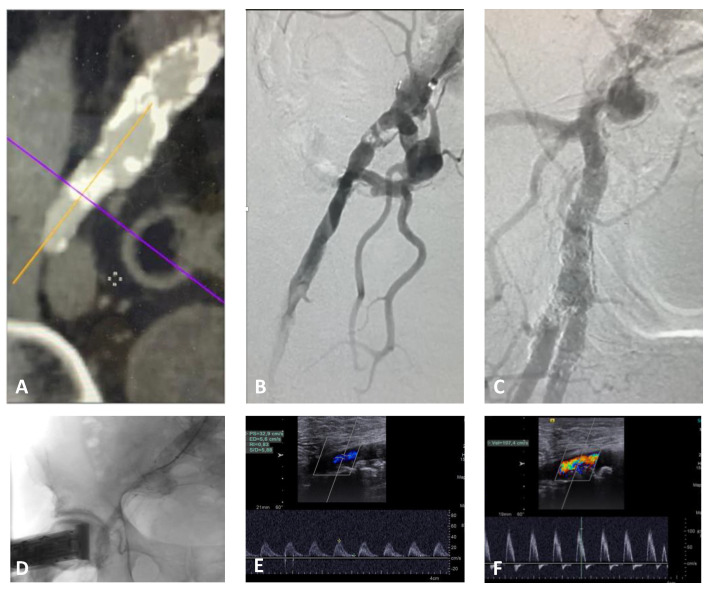
Case 1, right iliac external subocclusion, treated by intravascular lithotripsy without stenting from brachial access. (**A**) Preoperatory CT scan; (**B**) initial angiogram; (**C**) completion angiogram; (**D**) EVUS with preoperative (**E**) and postoperative (**F**) waveforms, changing from monophasic to biphasic flow.

**Figure 4 diagnostics-13-01356-f004:**
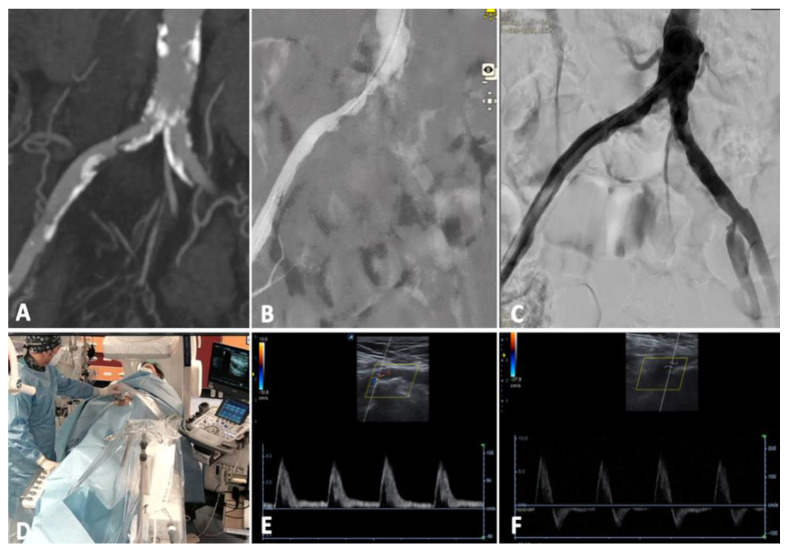
Case 2, right common iliac artery calcified stenosis treated by IVL stand-alone therapy from right femoral access. (**A**) Preoperatory CT scan.; (**B**) initial angiogram; (**C**) completion angiogram; (**D**) EVUS with preoperative (**E**) and postoperative (**F**) waveforms, changing from monophasic to biphasic flow.

**Figure 5 diagnostics-13-01356-f005:**
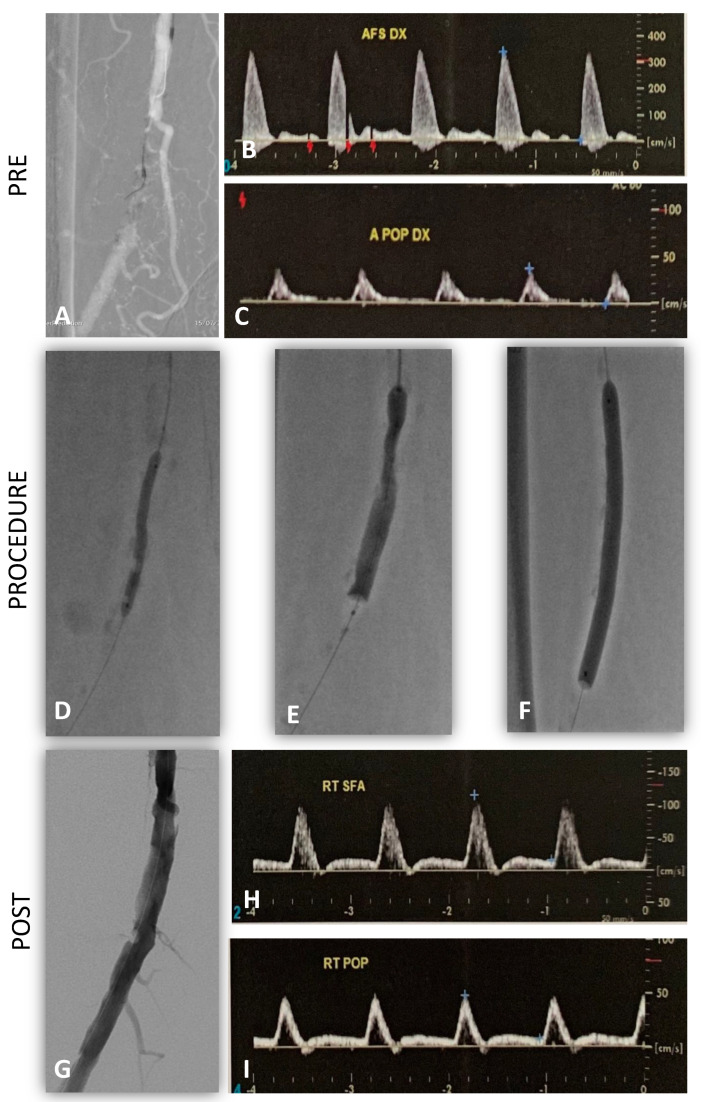
Case 3, right superficial femoral artery subocclusion treated by IVL/DCB. (**A**) Preoperative CA; (**B**) Preoperative DUS superficial femoral artery; (**C**) Preoperative DUS popliteal artery; (**D**) POBA predilatation; (**E**) IVL at first cycle; (**F**) DCB after IVL (10 cycles) treatments; (**G**) Postoperative CA; (**H**) Postprocedural DUS superficial femoral artery; (**I**) Postoprocedural DUS popliteal artery waveforms, changing from monophasic to triphasic flow.

**Figure 6 diagnostics-13-01356-f006:**
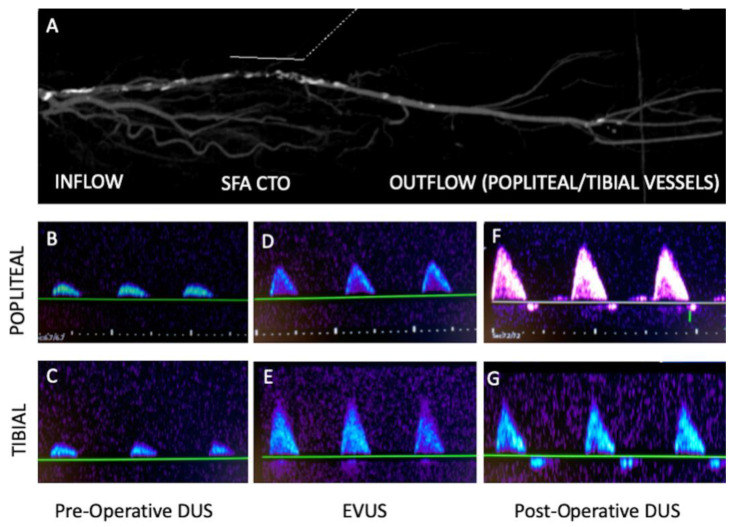
Case 4, right superficial femoral artery occlusion, treated by POBA/DCB. (**A**) CTA showing short SFA CTO with diseased inflow vessel (multiple stenoses with monophasic waveform) and patent outflow vessels/popliteal and tibial; (**B**,**C**) Preoperative DUS showing very low monophasic waveform at popliteal and tibial vessels. (**D**,**E**) EVUS shows a better outflow with improved monophasic waveforms. (**F**,**G**) Postoperative DUS showing a further better outflow with triphasic waveforms.

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
