# Peer review of "Extravascular Ultrasound (EVUS) to Assess the Results of Peripheral Endovascular Procedures"

_diagnostics, 2023, doi:10.3390/diagnostics13071356_

Round 1

Reviewer 1 Report

This article is about the use of “Extra Vascular UltraSound” (i.e. intra-operative duplex ultrasound) during peripheral endovascular revascularization procedures. Although DUS is widely used by many equipes as an intra-operative control during surgical revascularization, this is not the case for endovascular procedures, in which arteriography is the standard modality of control. Nonetheless, DUS adds some functional information, and therefore its use in this setting is worth reporting.

The article is overall well written. I have some requests of minor revisions:

1) In grading the severity of lower limb ischemia, the authors used the term “Rutherford Class”. I suggest to use the Rutherford category instead of class, as this therm may create ambiguity (please refer to Rutherford RB, Baker JD, Ernst C, Johnston KW, Porter JM, Ahn S, Jones DN. Recommended standards for reports dealing with lower extremity ischemia: revised version. J Vasc Surg. 1997 Sep;26(3):517-38. doi: 10.1016/s0741-5214(97)70045-4. Erratum in: J Vasc Surg 2001 Apr;33(4):805. PMID: 9308598.). For example in Case 1, line 126, “He referred 20 meters walking distance and nocturnal rest pain (Rutherford class II/III)”: usually this corresponds to Rutherford category 3/4; in line 156, the authors state that the patient in case 3 has 50 meters of walk intervals and a Rutherford class of II. Usually this walk interval corresponds to a Rutherford category of 3.

2) in line 58 the authors stated that the popliteal artery was explored through the popliteal fossa (i.e. posterior approach). However, in figure 1 it looks like the popliteal artery can be explored only from a medial approach (i.e. only in its proximal segment), unless the sterility of the operative field is lost, which is not advisable for an intra-operative control. Was the green sheet under the patient also sterile? Please clarify or replace the image. 

3) in line 234 you state that the cases presented were three, but they were four. Please correct.

Author Response

Reviewer 1

  1.  In grading the severity of lower limb ischemia, the authors used the term “Rutherford Class”. I suggest to use the Rutherford category instead of class, as this therm may create ambiguity (please refer to Rutherford RB, Baker JD, Ernst C, Johnston KW, Porter JM, Ahn S, Jones DN. Recommended standards for reports dealing with lower extremity ischemia: revised version. J Vasc Surg. 1997 Sep;26(3):517-38. doi: 10.1016/s0741-5214(97)70045-4. Erratum in: J Vasc Surg 2001 Apr;33(4):805. PMID: 9308598.). For example in Case 1, line 126, “He referred 20 meters walking distance and nocturnal rest pain (Rutherford class II/III)”: usually this corresponds to Rutherford category 3/4; in line 156, the authors state that the patient in case 3 has 50 meters of walk intervals and a Rutherford class of II. Usually this walk interval corresponds to a Rutherford category of 3.

Authors: thank you for your comments, we agree. The manuscript has been corrected according to reviewer’s request and reference has been added.

“He referred 20 meters walking distance and nocturnal rest pain (Rutherford category 3/4) and was previously treated by an endovascular abdominal aortic aneurysm repair (EVAR). Redline page 3 line 140.

“The patient was a 74 years old male, with a short (5 cm) calcific occlusion of the right superficial femoral artery. Walking range was <100 meters (Rutherford category 3). Redline page 4 line 204.

“The patient was a female, 72 years old with a short occlusion of the right superficial femoral artery. Patient had short (50 meters) walking range (Rutherford category 3). Redline page 5 line 254.

“Patient had short walking range (< 50 m), Rutherford category 3. Redline page 4 line 173.

Ref. Rutherford RB, Baker JD, Ernst C, Johnston KW, Porter JM, Ahn S, Jones DN. Recommended standards for reports dealing with lower extremity ischemia: revised version. J Vasc Surg. 1997 Sep;26(3):517-38. doi: 10.1016/s0741-5214(97)70045-4. Erratum in: J Vasc Surg 2001 Apr;33(4):805. PMID: 9308598. Redline page 8 line 871.

  1. in line 58 the authors stated that the popliteal artery was explored through the popliteal fossa (i.e. posterior approach). However, in figure 1 it looks like the popliteal artery can be explored only from a medial approach (i.e. only in its proximal segment), unless the sterility of the operative field is lost, which is not advisable for an intra-operative control. Was the green sheet under the patient also sterile? Please clarify or replace the image.

Authors: The green drape was sterile and allowed us to check popliteal flow without contamination of the field. We added specifications.

“The leg is prepared with a double sterile drape to approach any leg spots during the procedure.” Redline page 7 line 767.

  1. in line 234 you state that the cases presented were three, but they were four. Please correct.

Authors: Thank you for your comment. The cases have been divided in two sections (iliac and femoro-popliteal treatment), and presented in the discussion focusing on most important features, to highlight the benefits of using EVUS.

Reviewer 2 Report

I agree with authors that intraoperative duplex is useful to assess the immediate postprocedure flow. I have some questions and comments:

1. "Extra vascular" should be one word.

2. Protocol and Figure2: Stiff artery such as in dialysis patient may present with monophasic waveforms without significant stenosis. Also infected foot can cause hyperemic waveform changes and may become "monophasic". The waveforms with delayed/dampened upstroke waveforms definitely suggest the presence of proximal lesion.

2. L135: Why was the image unclear with CA? - Looking at the figure 3, I assume due to the calcium burden. Please describe it in the paragraph for a better understanding.

3. L147-8: What do authors mean by suboptimal vessel profile? Is it "flow unlimiting dissection"?

4. L154-L168: Please demonstrate the imaging. This case is only followed up for 1 month, and not very convincing.

5. L163: monophasic waveforms in inflow suggests aortoiliac occlusive disease. What are the findings?

6. L179: Duplex waveforms should not dictate the indication for DCB. In this Case 4, authours already checked the pressure gradient, and I do not see a point of using EVUS in this case.

Author Response

Reviewer 2

  1. "Extra vascular" should be one word.

Authors: Thank you for your comment. The article has been changed accordingly.

“ExtraVascular UltraSound (EVUS) to assess the results of peripheral endovascular procedures.” Redline page 1 line 1-3.

“By using a similar acronym, the intraoperative DUS has been previously described as ExtraVascular UltraSound (EVUS).”  Redline page 1 line 15-17.

“The EVUS (ExtraVascular UltraSound) protocol can be applied to patients undergoing endovascular repair of peripheral lesions.” Redline page 2 line 61-62.

“EVUS (ExtraVascular UltraSound): sterile setting in the operatory room before a peripheral endovascular procedure (occlusion of the right superficial femoral artery ) with three different spots upon the projections of femoral, popliteal and tibial arteries.” Redline Fig. 1 description

  1. Protocol and Figure2: Stiff artery such as in dialysis patient may present with monophasic waveforms without significant stenosis. Also infected foot can cause hyperemic waveform changes and may become "monophasic". The waveforms with delayed/dampened upstroke waveforms definitely suggest the presence of proximal lesion.

Authors: Thank you for your comment. We highlighted these adjunctive causes in the discussion.

“In case of significant stenosis or CTO, duplex scan usually shows a monophasic pattern of the pulsed doppler spectral analysis distal to the target lesion. Furthermore, a triphasic waveform is rarely detected in elderly patients with very diffuse atherosclerotic disease, dyalisis or cases with previous procedure (eg. Stenting), due to the lack of elasticity in vessel walls. In such cases, a preoperative mono-phasic waveform progressing into a bi-phasic postoperative waveform should represent a technical success.”  Redline page 2 line 86-90.

The arterial stiffness in some patients (dialysis, long stenting, diffused calcifications, narrowed vessels) may present with monophasic waveforms without significant stenosis.
Redline page 6 line 591-592

  1. L135: Why was the image unclear with CA? - Looking at the figure 3, I assume due to the calcium burden. Please describe it in the paragraph for a better understanding.

Authors: Thank you for your comment. The information has been added to the manuscript.

“Due to the extensive vessel calcifications, the completion angiogram showed an unclear imaging (good flow in term of contrast medium velocity with a suboptimal arterial vessel filling) despite the use of two different C-arm projections. Redline page 4 line 165-167.

  1. L147-8: What do authors mean by suboptimal vessel profile? Is it "flow unlimiting dissection"?

Authors: Completion angiogram showed some walls irregularities.

After the procedure, luminal gain was completely restored, and the CA showed a vessel profile with few irregularities and very good flow. Redline page 4 line 211-212.

  1. L154-L168: Please demonstrate the imaging. This case is only followed up for 1 month, and not very convincing.

Authors: Thank you for your comment. We have the duplex imaging and therefore we added another figure (n°6) and more specifications about the case.

The preoperative DUS showed a direct/suboptimal inflow (proximal superficial femoral artery with good monophasic waveform, sign of a small vessel with multiple stenosis) and a poor outflow (popliteal artery with a very low monophasic waveform). [Fig. 6] Redline page 5 line 264-267.

The first post-operative day, thanks to the better compliance and the vessel remodelling after the recanalization, the waveforms showed a further improvement with triphasic flow in popliteal and tibial artery. [Fig. 6] Redline page 5 line 274-276.

  1. L163: monophasic waveforms in inflow suggests aortoiliac occlusive disease. What are the findings?

Authors: Patient had diffuse disease in proximal superficial femoral artery (the occlusion was distal) with multiple non flow-limiting lesions which caused the worsening of arterial stiffness and resulted in a low monophasic waveform, worsened by a diffused narrow diameter. This was the main reason to avoid further treatment after the recanalization (see previous response)

  1. L179: Duplex waveforms should not dictate the indication for DCB. In this Case 4, authors already checked the pressure gradient, and I do not see a point of using EVUS in this case.

Authors: Agreed. EVUS was used to crosscheck its usefulness in presence of other available methods, such as pressure gradient that is feasible and useful as recommended by TASC II for iliac lesions. As described, the setup for EVUS is ready from the beginning and is not time-consuming, above all in this district.  Moreover, in this case we are describing a not-standardized procedure (IVL stand-alone therapy in place of stenting). Therefore, we believe that having different methods to asses our best result could be useful from many points of view.

Reviewer 3 Report

This is a nice, well-written proof of concept paper for EVUS utility in the assessment of peripheral artery status in patients undergoing endovascular procedures. 

Author Response

Reviewer 3

This is a nice, well-written proof of concept paper for EVUS utility in the assessment of peripheral artery status in patients undergoing endovascular procedures.

Authors: Thank you for your comment.